# Nerve-Sparing Systematic Lymph Node Dissection in Gynaecological Oncology: An Innovative Neuro-Anatomical and Surgical Protocol for Enhanced Functional Outcomes

**DOI:** 10.3390/cancers12113473

**Published:** 2020-11-22

**Authors:** Mustafa Zelal Muallem, Yasser Diab, Thomas Jöns, Jalid Sehouli, Jumana Muallem

**Affiliations:** 1Department of Gynecology with Center for Oncological Surgery, Corporate Member of Freie Universität Berlin, Humboldt-Universität zu Berlin, and Berlin Institute of Health, Berlin, Virchow Campus Clinic, Charité Medical University, 13353 Berlin, Germany; Jalid.sehouli@charite.de (J.S.); jumana.muallem@charite.de (J.M.); 2Department of Gynecology, Portland Hospital, Portland, 3305 VIC, Australia; ydiab.pdh@swarh.vic.gov.au; 3Department of Anatomy, Mitte Campus Clinic, Charité Medical University, 10117 Berlin, Germany; Thomas.joens@charite.de

**Keywords:** nerve-sparing lymphadenectomy, pelvic and para-aortic lymph nodes, lymph node dissection, infra-renal bilateral template, aortic plexus

## Abstract

**Simple Summary:**

One of the most frequent complications of the systematic lymph node dissection (SLND) is the injury of autonomic nervous system in the para-aortal region during the procedure. These injuries are supposed to be responsible for some of the postoperative bladder, bowel, and sexual dysfunctions. The poor anatomical understanding of the sympathetic nerves within the boundaries of an infra-renal bilateral template has limited the promulgation of a precise nerve-sparing surgery during such SLND. Therefore, the principal goal of the present study was to provide the first ever-comprehensive exposition of the anatomy of the female aortic plexus and superior hypogastric plexus and their variations. This exposition was achieved by strategic dissection of 19 human female cadavers and extrapolating the findings to develop a precise surgical technique for more accurate navigation into these structures during nerve-sparing SLND in 15 cervical cancer patients and 48 ovarian cancer patients.

**Abstract:**

Whilst systematic lymph node dissection has been less prevalent in gynaecological cancer cases in the last few years, there is still a good number of cases that mandate a systematic lymph node dissection for diagnostic and therapeutic purposes. In all of these cases, it is crucial to perform the procedure as a nerve-sparing technique with utmost exactitude, which can be achieved optimally only by isolating and sparing all components of the aortic plexus and superior hypogastric plexus. To meet this purpose, it is essential to provide a comprehensive characterization of the specific anatomy of the human female aortic plexus and its variations. The anatomic dissections of two fresh and 17 formalin-fixed female cadavers were utilized to study, understand, and decipher the hitherto ambiguously annotated anatomy of the autonomic nervous system in the retroperitoneal para-aortic region. This study describes the precise anatomy of aortic and superior hypogastric plexus and provides the surgical maneuvers to dissect, highlight, and spare them during systematic lymph node dissection for gynaecological malignancies. The study also confirms the utility and feasibility of this surgery in gynaecological oncology.

## 1. Introduction

Dissection evaluation of pelvic and para-aortic lymph nodes has been an integral component of the surgical staging protocol for several gynaecologic malignancies for over a century [1,2]. However, in the recent years, this systematic dissection evaluation of lymph nodes has become less relevant because of the extensive deployment of sentinel lymph node mapping technology with a corroborated efficacy in the detection of several endometrial and cervical cancers [3]. Further, the recently published results of the randomized trial on lymphadenectomy in patients with advanced ovarian neoplasms (LION trial) [4] have narrowed down the scope for systematic lymph node dissection and, in fact, restricted the procedure only to apparent early stage ovarian cancer patients [5].

Nevertheless, the significance of the prognostic value of the para-aortic lymph node status in locally advanced cervical cancer has been accentuated again in the new International Federation of Gynaecology and Obstetrics (FIGO) staging system for cervical cancer that classifies patients with a para-aortic lymph node involvement in stage FIGO IIIC2 [6].

These developments in surgical management of gynaecological malignancies have given rise to austere diagnostic restrictions for a systematic lymph node dissection in such cases. Taking these facts and the potential additional treatment burden of a systematic lymph node dissection into account, all contentions are in place to avoid the complications of a lymph node dissection. One of the most frequent and vital complications of the para-aortic lymph node dissection that has perhaps never descended into the cognitive focus of many gynaecologic oncologists is the injury of autonomic nervous system in the para-aortal region during the procedure. The present study supposes that these nerve injuries are responsible for some of the postoperative bladder, bowel, and sexual dysfunctions.

The present study also contends that a poor anatomical understanding of the sympathetic nerves within the boundaries of an infra-renal bilateral template has limited the promulgation of a precise nerve-sparing surgery during such systematic lymph node dissections. Since we already described the precise anatomy of pelvic autonomic nervous system (inferior hypogastric plexus) in our previous studies [7,8,9], the principal goal of the present study was to provide the first ever-comprehensive exposition of the anatomy of the female aortic plexus and superior hypogastric plexus and their variations, to elucidate and improve the surgical outcomes of a systematic nerve-sparing lymph node dissection.

This exposition was achieved by strategic dissection of human female cadavers and extrapolating the findings to develop a precise surgical technique for more accurate navigation into these structures during nerve-sparing systematic lymph node dissections, especially in the para-aortic region.

## 2. Results

### 2.1. The Anatomy of Aortic Plexus

The abdominal aortic plexus is the sympathetic network of autonomic nerves overlying in the front and sides of the abdominal aorta in the infra-renal bilateral template and going along with the superior hypogastric plexus at the sacral promontory [10,11]. The superior hypogastric plexus, often called the presacral nerve, terminates by dividing into right and left hypogastric nerves and consists primarily of sympathetic and visceral afferent fibers [12]. These fibers merge with the pelvic splanchnic nerves (parasympathetic) in the deep extraperitoneal spaces at the lateral sidewall of the vagina and rectum, to form the inferior hypogastric plexus bilaterally from which nerve fibers spread out to the pelvic organs [7,13,14,15]. The aortic plexus receives sympathetic supply from the lumbar sympathetic trunk and suprarenal pre-aortic plexuses via lumbar splanchnic nerves and intermesenteric nerves, respectively [16]. The new technique in this study effectively comprises a precise description of these nerve courses for an accurate dissection that incorporates precise nerve-sparing practices during such bilateral para-aortic lymph node dissections and efficaciously distinguishes them from the most caudal parts, to their origin, from the sympathetic trunks bilaterally.

### 2.2. Dissection of the Right Cord of the Aortic Plexus

The cadaver study has primarily shown that the easiest way to identify the lumbar splanchnic nerves is to prepare and dissect them from their caudal most part at the superior hypogastric plexus in the presacral area (the point of bifurcation of the common iliac arteries) and then to track carefully the deepest lumbar splanchnic nerve (which was, in all of our cases, the inferior right lumbar splanchnic nerve), to its point of origin at the second lumbar ganglion of the right sympathetic trunk. 

The study has also elucidated that it would be easy, in most cases (81.7%, 67/82 cases), to identify a ganglion at the right side of the aorta around the inferior mesenteric artery or maximal 1 cm caudal from it (the inferior mesenteric ganglion). Successful identification of this ganglion allows the dissection to go laterocranial and dorsal to the vertebral column in the interaortocaval space above the middle lumbar vein and then behind the vena cava, at the level of the right lumbar artery (Figure 1 shows the right cord of aortic plexus and its components).

Dissecting along this nerve behind the cava and then laterally gives the surgeon a comprehensive prospect to identify the right sympathetic chain and the origin of the inferior lumbar splanchnic nerve from the second lumbar ganglion of the right sympathetic chain. An accessory inferior right lumbar splanchnic nerve was identified in 72% of the maneuvers (59/82) in the study. This accessory nerve originates in our cases from the third lumbar ganglion of the right sympathetic chain and joins the inferior right lumbar splanchnic nerve before reaching the inferior mesenteric ganglion.

The superior right splanchnic nerve will be easy to identify when dissecting along to its origin from the first lumbar ganglion of the right sympathetic chain behind and lateral of vena cava at the level where the left renal vein inserts into vena cava crossing over the right superior lumbar vein. (Figure 2 shows the emergence of superior and inferior lumbar splanchnic nerves in the first and second lumbar ganglions directly at the level of superior and middle lumbar veins in one of the cadaver preparations in this study.)

In 18.3% of the cases in the study, two ganglions at the right and ventral side of the aorta, instead of the one big inferior mesenteric ganglion, were identified. Further, a ganglion at the level of inferior mesenteric artery but lateral at the right side, where only the inferior lumbar splanchnic nerve crosses, could also be further highlighted. (The authors term it the prehypogastric ganglion, as referred to in the case of most anatomical studies of male cadavers). In all of these cases, the superior lumbar splanchnic nerve is identified as crossing another even smaller ganglion directly caudal from the origin of right ovarian artery from the aorta. (The authors term it the ovarian or gonadal ganglion).

By resecting the lymph node in the paracaval region, there is truly little or no chance to injure the right lumbar splanchnic nerves or the right sympathetic chain during the surgery. The dissection and resection of lymph nodes in the intra-aortocaval region, on the other hand, always leads to an injury to the right lumbar splanchnic nerves, the inferior mesenteric ganglion, the prehypogastric ganglion, and the ovarian (gonadal) ganglion, if the surgeon does not pay sufficient attention to recognize, highlight, and carefully isolate these nervous components (Figure 3 shows a medial and lateral view of the dissected right cord of the aortic plexus).

### 2.3. Dissection of the Left Cord of the Aortic Plexus

On the left side, the inferior left lumbar splanchnic nerve can be easily identified when dissected in its short course between the superior hypogastric plexus and the left inferior mesenteric ganglion at the lateral left sidewall of the aorta around the origin of inferior mesenteric artery. The inferior left lumbar splanchnic nerve could then be followed in a dorsocranial short course to its origin from the second lumbar ganglion of the left sympathetic trunk, at a level where the left lumbar artery and vein disappear deep into psoas major. This level is placed more superiorly than its right-sided counterpart, the inferior right lumbar splanchnic nerve (Figure 4 shows a lateral view of the dissected left cord of the aortic plexus).

The superior left lumbar splanchnic nerve can be identified when dissected following its course from the upper part of left inferior mesenteric ganglion to its origin from the first lumbar ganglion of the left sympathetic chain directly dorsal to the inferior border of the left renal vein at the level of the renal–lumbar vein. This vein could be emphatically identified in 59.7% of the maneuvers in the present study (49/82), and it obviously complicated the clear exposure of the first lumbar ganglion of the left sympathetic chain (Figure 5 shows a lateral view of the dissected left cord of the aortic plexus, with a focus on the complex anatomy at the origin of the superior left lumbar splanchnic nerve).

## 3. Discussion

Whilst the systematic lymph node dissection is less prevalent in the gynaecological cancer cases in the last few years for the reasons stated and expounded in the introduction, there is still a good number of cases that mandate a systematic lymph node dissection for diagnostic purposes especially, for those in the early stages of ovarian cancer. In such cases, the status of the node is vital for numerous reasons. A systematic lymph node dissection primarily allows for an accurate assessment of the clinical stage, which often influences the medical decision for an adjuvant therapy that might bring about a complete cytoreduction of the cancer [17]. This is diagnostically germane even in the absence of published randomized controlled trials to substantiate the efficacy of lymphadenectomy. 

Moreover, the presence of affected para-aortic lymph nodes at the time of the primary diagnosis of a locally advanced cervical cancer correlates closely with a shorter disease-free survival [18] and worsened overall survival [19]. It is likewise imperative to note that, in spite of the availability of advanced imaging and mapping techniques, several studies have confirmed the presence of para-aortic lymph node metastases in 8–12% of the cases after surgical staging in the para-aortic region despite negative results with diagnostic imaging techniques [18,20,21,22]. For these reasons, it is crucial to acknowledge the diagnostic value of systematic lymph node dissection in locally advanced cervical cancers. Furthermore, the randomized Uterus-11 Trial has revealed an improved scope for disease-free and overall survival after surgical staging as compared to CT-guided staging, which could be interpreted as corroboration of a minor therapeutic advantage with the systematic lymph node dissection in this cancer cohort [23]. 

In all of these cases and others where the systematic lymph node dissection will be rather strongly indicated, it is crucial to perform the procedure as a nerve-sparing technique with utmost exactitude, which can be achieved optimally only by isolating and sparing the all components of the aortic plexus. Failure in the isolation and sparing of all the components of aortic plexus has been established to lead to numerous postoperative urinary, bowel, and sexual functional disorders. In such a context, the clinical legitimacy of a nerve-sparing para-aortal lymph node dissection protocol is severely limited by the lack of published studies that efficaciously integrate the aforementioned anatomy and corroborate the relationships with one another by pertinent examination and explication.

One of the most noticeable complications after a surgical injury of the aortic plexus and/or superior hypogastric plexus (sympathetic nervous system) during systematic lymph node dissection is the postoperative impairment of the sexual function. Impairment of the sexual function is, in fact, the most enduringly compromised quality-of-life (QOL) issue encountered after treatment for gynaecologic cancers, affecting up to 50% of the patients [24,25,26]. The most persistent and negative repercussions have been the loss of patients’ libido and a diminished vaginal lubrication [27]. This is primarily due to the fact that the vaginal blood-flow response to a sexual stimulus and subsequent vaginal lubrication are obvious neural reflex responses innervated by the hypogastric nerve (sympathetic innervation) [28,29,30,31,32]. There is growing evidence to show that the sympathetic nervous system activation plays a facilitatory role in women’s physiological sexual arousal [33] and diminishing its activity state hinders the subjective sexual arousal in sexually functional women [34].

In the recently published data of a sub-protocol of the prospectively randomized LION trial, comparing two cohorts that differed only in the performance of a lymphadenectomy, the role of radical surgery in the retroperitoneal space has been prospectively substantially evaluated with reference to the sexual function (sub-study LION-PAW). It is imperative to mention that, in the LION trial, there have been no requirements for nerve-sparing surgical techniques [35]. The sub-study LION-PAW study has confirmed that lymphadenectomy (unsparing technique) is associated with a higher incidence of post-operative complications, including a significant change of the orgasm score from baseline to 12 months in sexually active patients. This result falls in tandem with the still unpublished findings by the authors on sexual function assessment of advanced ovarian cancer patients [36], who were operated by conventional surgical technique compared with the patients operated with total retroperitoneal en bloc resection of multivisceral-peritoneal packet (TROMP) technique [37] and abovementioned nerve-sparing lymph node dissection. The data have shown a worsened score of discomfort during sexual intercourse and a reduced orgasm score with conventional surgical technique as compared to nerve-sparing lymph node dissection. However, the clinical expediency of such a nerve-sparing approach is still limited due to a lack of published studies, integrating the aforementioned anatomy in control randomized trials to compare the functional outcomes after nerve-sparing lymph node dissection with the conventional systematic nerve-insulting lymphadenectomy procedure.

## 4. Materials and Methods

The anatomic dissections of two fresh and 17 formalin-fixed female cadavers were utilized to study, understand, and decipher the hitherto ambiguously annotated anatomy of the autonomic nervous system in the retroperitoneal para-aortic region. Rigorous dissection protocol was in place to interpret the retroperitoneal para-aortic nervous connections to and from the contiguous anatomical structures, with specific reference to lymph node dissection in gynae-oncology. The new anatomical know-how from this cadaver study was utilized to enhance and develop a superior technique for para-aortic lymph node dissection and was efficaciously employed in locally advanced cervical cancer (a cohort of 15) [9] and in advanced ovarian cancer as a part of total retroperitoneal en bloc resection of multivisceral-peritoneal packet (TROMP operation) for advanced ovarian cancer (a cohort of 48) [37,38] and the evidence appositely reported in multiple journals. Approval from the Charité Local Ethics Committee was provided for the clinical parts of this study (EA1/174/14 for cervical cancer patients and EK207/2003 Amendment 15/2012 for ovarian cancer patients) as per the recommendations of the Ethics Commission of the parent institution. The data from anatomic dissections on cadavers were obtained in accordance with Charité guidelines for cadaveric use in research. 

We aimed in this study to describe the whole components of abdominal aortic plexus anatomically and to develop a surgical technique to help us to spare them during systematic lymph node dissection (The video presents the technique of laparoscopic nerve-sparing lymph node dissection [39]). The following papers will focus on the functional outcomes of sparing the abdominal aortic plexus during systematic lymph node dissection for gynecological cancers.

## 5. Conclusions

The nerve-sparing systematic lymph node dissection is feasible in gynaecological malignancies by following the aforementioned anatomical depended surgical technique for dissection the aortic and superior hypogastric plexus. This might enhance the postoperative functional outcomes, especially the sexual complications after such surgeries.

## Figures and Tables

**Figure 1 cancers-12-03473-f001:**
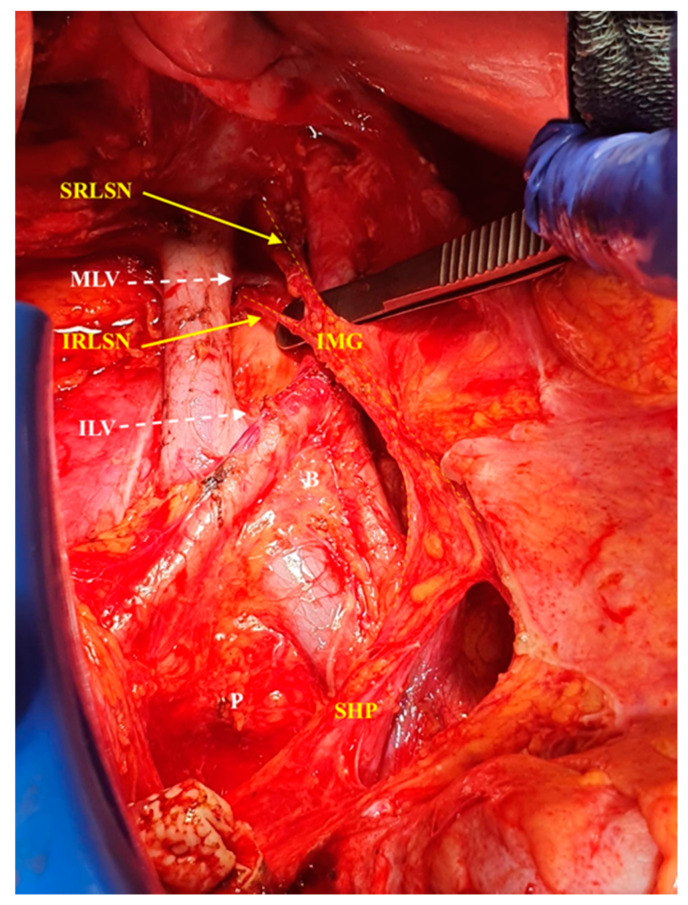
The right cord of aortic plexus. SRLSN, superior right lumbar splanchnic nerve; MLV, middle lumbar vein; IRLSN, inferior right lumbar splanchnic nerve; ILV, inferior lumbar vein; IMG, inferior mesenteric ganglion; SHP, superior hypogastric plexus.

**Figure 2 cancers-12-03473-f002:**
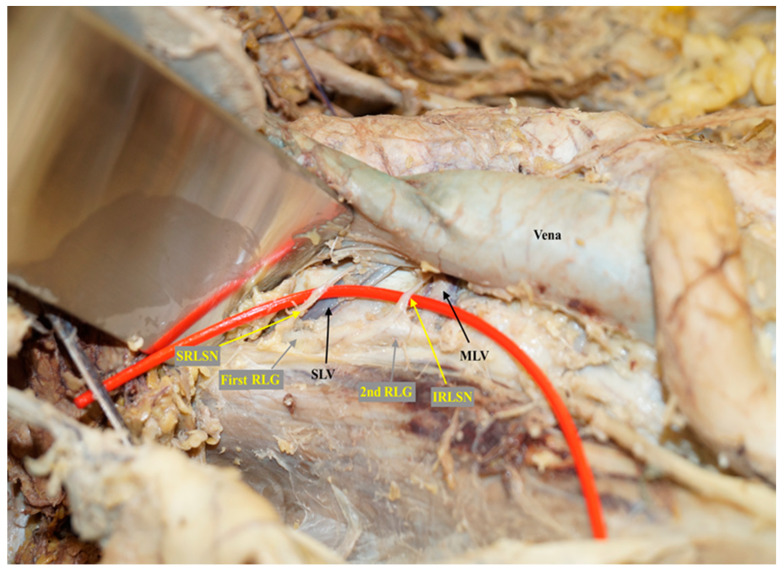
The emerging of superior and inferior lumbar splanchnic nerves in the first and second lumbar ganglions directly at the level of superior and middle lumbar veins in one cadaver preparation for this study. SRLSN, superior right lumbar splanchnic nerve; SLV, superior lumbar vein; 1st RLG, first right lumbar ganglion of sympathetic trunk; IRLSN, inferior right lumbar splanchnic nerve; MLV, middle lumbar vein; 2nd RLG, second right lumbar ganglion of sympathetic trunk.

**Figure 3 cancers-12-03473-f003:**
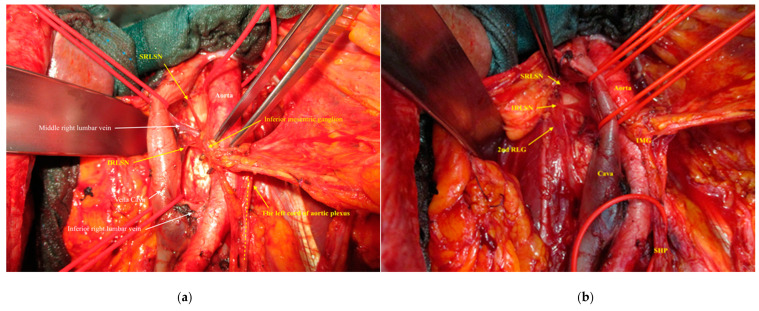
Medial and lateral views of the dissected right cord of the aortic plexus in a patient during the primary cytoreductive surgery for ovarian cancer. (**a**) Medial view (interaortocaval space): SRLSN, superior right lumbar splanchnic nerve; IRLSN, inferior right lumbar splanchnic nerve. (**b**) Lateral view (para- and retrocaval space): SRLSN, superior right lumbar splanchnic nerve; IRLSN, inferior right lumbar splanchnic nerve; 2nd RLG, second right lumbar ganglion of sympathetic trunk; IMG, inferior mesenteric ganglion; SHP, superior hypogastric plexus.

**Figure 4 cancers-12-03473-f004:**
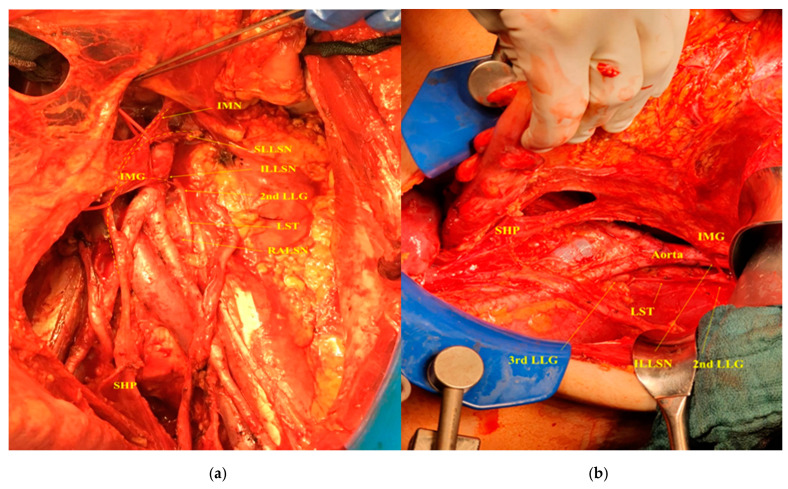
(**a**) Ventral view and (**b**) lateral view of the dissected left cord of the aortic plexus in a patient during the primary cytoreductive surgery for ovarian cancer. IMN, intermesenteric nerve; SLLSN, superior left lumbar splanchnic nerve; ILLSN, inferior left lumbar splanchnic nerve; 2nd LLG, left lumbar ganglion; LST, left sympathetic trunk; RALSN, retroaortic lumbar splanchnic nerve; IMG, inferior mesenteric ganglion; SHP, superior hypogastric plexus.

**Figure 5 cancers-12-03473-f005:**
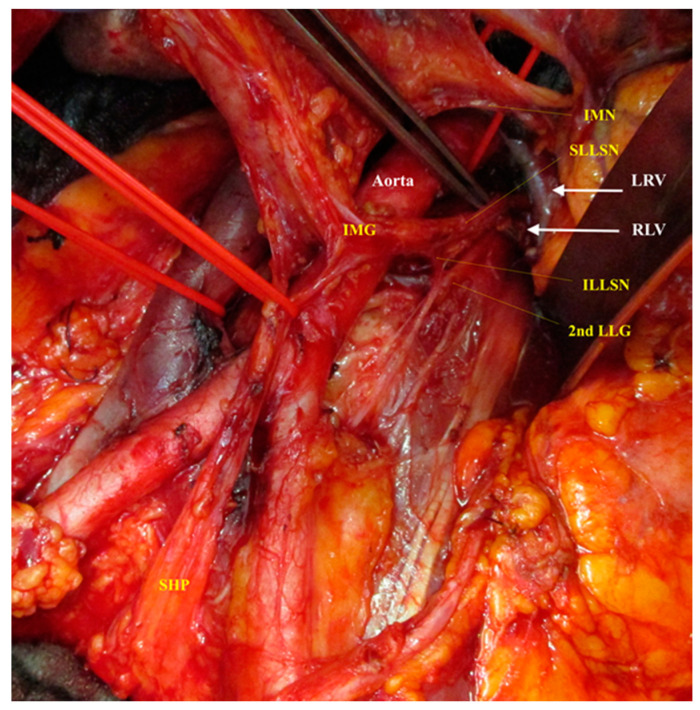
A lateral view of the dissected left cord of the aortic plexus with focusing on the complex anatomy at the origin of superior left lumbar splanchnic nerve. IMN, intermesenteric nerve; SLLSN, superior left lumbar splanchnic nerve; LRV, left renal vein; RLV, renal–lumbar vein; ILLSN, inferior left lumbar splanchnic nerve; 2nd LLG, left lumbar ganglion; IMG, inferior mesenteric ganglion; SHP, superior hypogastric plexus.

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
