# Peer review of "Nerve-Sparing Systematic Lymph Node Dissection in Gynaecological Oncology: An Innovative Neuro-Anatomical and Surgical Protocol for Enhanced Functional Outcomes"

_cancers, 2020, doi:10.3390/cancers12113473_

Round 1

Reviewer 1 Report

Very interesting study.

I would suggest the authors to provide some anatomical drawings in order to improve the paper.

Author Response

Thank you very much for reviewing our manuscript and for your valuable comments.

We add an anatomical drawing as a graphical abstract to present all the component of aortic plexus. 

Reviewer 2 Report

I think it is very well written and the pictures are very clear. One comment that I have is that the positions of the lumbar nerves are prone to variation and are not bilaterally symmetrical. This is applicable also to ganglia as well. So it is really difficult to define a specific location for these plexus based on few cadavers and also few patients. 

Also If they  can add a section that describes the functions of these plexus (evidence based with good references) that would be helpful for the readers.

Also; in Line 28;  please change too. I believe it meant to be  to   and not  too.

Author Response

Thank you very much for reviewing our manuscript and for your valuable comments.

  • I think it is very well written and the pictures are very clear. One comment that I have is that the positions of the lumbar nerves are prone to variation and are not bilaterally symmetrical.
  • That is right and we describe this even clearly in our paper.

  • This is applicable also to ganglia as well. So it is really difficult to define a specific location for these plexus based on few cadavers and also few patients.
  • You are very right, that there are a lot of variations. The most common ones were described in our paper. Our collective consists of 19 cadavers, 15 cervical cancer patients and 48 ovarian cancer patients (totally 82 patients, which are definitely more than few patients). However, we still think that there are some other undescribed variations, but we believe that following this anatomical description and the surgical steps mentioned in this study will help surgeons to spare the most important components of aortic plexus in the most of patients.

  • Also If they  can add a section that describes the functions of these plexus (evidence based with good references) that would be helpful for the readers.
  • In discussion we pointed some studies up, which concentrated on the functions of aortic plexus specially in women. References 30 to 36.

  • Also; in Line 28;  please change too. I believe it meant to be  to   and not  too.
  • Actually, we meant (too).

Reviewer 3 Report

In this study, anatomic dissection was performed on 2 fresh and 17 formalin-fixed cadavers to clarify the anatomy of the autonomic nervous system in the retroperitoneal para-aortic region useful for optimal lymph node dissection in gynecologic surgery, and the results were applied in actual gynecological cancer cases. This is an interesting article, and would be worth publication after revising the following points;

  1. Please add a schematic drawing of the anatomy of the autonomic nervous system in the retroperitoneal region clarified by the study.

  1. Please describe the methods of anatomical dissection more precisely; how fresh were the fresh cadavers? How to identify and confirm the autonomic nervous system in the cadavers? How to differentiate the nervous system from other structures?

  1. Please clearly describe the primary/secondary outcomes of the study on the clinical cases.

Author Response

Thank you very much for reviewing our manuscript and for your valuable comments.

  • Please add a schematic drawing of the anatomy of the autonomic nervous system in the retroperitoneal region clarified by the study.
  • This is a very helpful and valuable suggestion. We add a schematic drawing as a graphical abstract.

  • Please describe the methods of anatomical dissection more precisely; how fresh were the fresh cadavers? How to identify and confirm the autonomic nervous system in the cadavers? How to differentiate the nervous system from other structures?
  • The fresh cadavers is a term to describe cadavers, which are not exsanguinated or fixed with Formalin but are simply washed with antiseptic soap before being frozen at -20°C within a week of procurement. Approximately 3 days prior to use the cadaver is thawed at room temperature. These where used to try to dissect the aortic plexus laparoscopically as it is difficult to apply CO2-insufflation to formalin-fixed cadavers. The way to identify the nerves in cadavers and during surgery is exactly the same and that what we described in this paper. Identifying the whole courses of these nerves from sympathetic trunks to the ganglions will confirm us the integrity of these structures. Of course, other authors could use a histological studies to confirm the autonomic nerves but we abandoned that because our work aimed to spare these structures not to resect them to confirm their histology.

  • Please clearly describe the primary/secondary outcomes of the study on the clinical cases.
  • We aimed primary to describe the whole components of abdominal aortic plexus anatomically and to develop a surgical technique to help us to spare them during systematic lymph node dissection. This study could meet these purposes.
    Our secondary aims were to show the functional results of sparing the components of abdominal aortic plexus, which is the subject of our next paper. In this coming-soon paper, we will focus on the sexual functions that the affected bowel function will be difficult to be assessed after such big surgeries (primary cytoreductive surgery for ovarian cancer) with the following chemotherapy.
  • We added this to the methods.

Round 2

Reviewer 3 Report

Well revised manuscript, worth publication